# Propidium Monoazide (PMAxx)-Recombinase Polymerase Amplification Exo (RPA Exo) Assay for Rapid Detection of *Burkholderia cepacia* Complex in Chlorhexidine Gluconate (CHX) and Benzalkonium Chloride (BZK) Solutions

**DOI:** 10.3390/microorganisms11061401

**Published:** 2023-05-26

**Authors:** Soumana Daddy Gaoh, Ohgew Kweon, Youngbeom Ahn

**Affiliations:** Division of Microbiology, National Center for Toxicological Research, U.S. Food and Drug Administration, Jefferson, AR 72079, USA; soumana.daddy-gaoh@fda.hhs.gov (S.D.G.); oh-gew.kweon@fda.hhs.gov (O.K.)

**Keywords:** *Burkholderia cepacia* complex, recombinase polymerase amplification exo (RPA exo), propidium monoazide (PMAxx), non-sterilized pharmaceutical products

## Abstract

Both sterile and non-sterile pharmaceutical products, which include antiseptics, have been recalled due to *Burkholderia cepacia* complex (BCC) contamination. Therefore, minimizing the frequency of outbreaks may be conducive to the development of a quick and sensitive approach that can distinguish between live and dead loads of BCC. We have assessed an exo probe-based recombinase polymerase amplification (RPA) with 10 µM propidium monoazide (PMAxx) for selective detection of live/dead BCC cells in various concentrations of antiseptics (i.e., chlorhexidine gluconate (CHX) and benzalkonium chloride (BZK) solutions) after 24 h. The optimized assay conducted using a set of primer–probes targeting *gbpT* was performed at 40 °C for 20 min and shows a detection limit of 10 pg/µL of genomic DNA from *B. cenocepacia* J2315, equivalent to 10^4^ colony-forming units (CFU/mL). The specificity of a newly designed primer and probe was 80% (20 negatives out of 25). The readings for total cells (i.e., without PMAxx) from 200 µg/mL CHX using PMAxx-RPA exo assay was 310 relative fluorescence units (RFU), compared to 129 RFU with PMAxx (i.e., live cells). Furthermore, in 50–500 µg/mL BZK-treated cells, a difference in the detection rate was observed between the PMAxx-RPA exo assay in live cells (130.4–459.3 RFU) and total cells (207.82–684.5 RFU). This study shows that the PMAxx-RPA exo assay appears to be a valid tool for the simple, rapid and presumptive detection of live BCC cells in antiseptics, thereby ensuring the quality and safety of pharmaceutical products.

## 1. Introduction

The genus *Burkholderia* comprises over 120 species of Gram-negative bacteria found in a variety of ecological niches [1,2,3]. Specifically, strains of the *Burkholderia cepacia* complex, or BCC, consist of a group of bacteria that have recently been linked to numerous outbreaks. This resulted in contamination of disinfectants, intravenous solutions, nebulizer solutions, mouthwash products, medical devices, respiratory treatment equipment as well as antiseptics (e.g., chlorhexidine gluconate (CHX) and benzalkonium chloride (BZK)) [1,2]. Moreover, during the past decade BCC has emerged as the leading contaminant in non-sterile water-based drugs and non-drug products, resulting in numerous recalls [4]. According to an FDA report released in 2019, *Burkholderia* spp. was the most common cause of non-sterile drug recalls (105 recalls), followed by *Ralstonia pickettii* (45 recalls), and *Salmonella* spp. (28 recalls) [5]. BCC organisms are common in water-based environments, where they can thrive in low nutrient conditions. They can also persist despite the use of antimicrobial preservative systems and antiseptics, posing a major threat to many susceptible individuals [4,5]. The United States Food and Drug Administration (FDA) was justifiably concerned with BCC contamination of non-sterile drug products in 2017, issuing an advisory notice. Subsequently, following up on a request from a stakeholder, the United States Pharmacopeia (USP) published Microbiological Examination of Non-sterile Products: Tests for *Burkholderia cepacia* complex [6], which contains test procedures and media formulations for BCC detection to guarantee that the quality of drug ingredients, pharmaceutical-grade water, and finished drug products conform to appropriate standards. These tests rely on culture methods and biochemical identification, are time-consuming and are limited by their low specificity and sensitivity. Therefore, a rapid, reliable and cost-effective method for the detection of BCC in pharmaceutical settings is vital to ensure public safety.

Although molecular techniques—based on polymerase chain reaction (PCR)—have been widely used to detect BCC, their reliance on complex heat cycling devices, trained workers and a time-consuming workflow has limited their application for on-site testing or in resource-constrained situations [7,8]. A promising alternative to PCR, isothermal amplification of nucleic acids, achieves quick and effective amplification at a constant temperature, without the need for thermocycling [9,10,11]. Methods such as loop-mediated isothermal amplification (LAMP) or recombinase polymerase amplification (RPA) have become widely available applications. The need for a single temperature incubation reduces equipment requirements and the elimination of repeated heating and cooling steps provides faster reaction times, making them very attractive for implementation in low-resource settings. Recently, we developed a LAMP assay for detecting BCC in non-sterile pharmaceutical products [12]. Our LAMP assay was successful in detecting as low as 1.034 pg/μL of genomic DNA, with a 10 μL reaction volume in 25 min at 65 °C. The LAMP assay is a very specific, simple, fast and cost-effective tool, which makes live BCC detection in water-based matrices a possibility. The LAMP assay relies on a complex primer design (i.e., set of six primers) at an optimal temperature of 65 °C, which can constrain target site selection, resolution, and/or specificity. Although the LAMP assay is a sensitive and specific method, it is a limited field-based application PCR to presumptively identify BCC. An isothermal gene amplification method offers an alternative for the rapid detection of pathogens in field-based, resource-limited areas or point-of-care diagnostics [11]. 

RPA reactions are relatively easy to perform with a simple, portable, battery-powered device, and do not require complex equipment or extensive training. The mechanism involves two major proteins to substitute for the traditional heat denaturation step in PCR: the *Escherichia coli* recombinase A (*recA*) and single-stranded DNA binding protein (SSB) to match primers to their target on the template DNA [9,11]. Then, the *Staphylococcus aureus* (Sau) polymerase extends the primer and displaces the native strand. Real-time monitoring detection of fluorescence DNA amplification can be achieved using exo probes (TwistDX, Cambridge, UK). The reaction is rapidly carried out in a fluorometer with repeated cycles resulting in the exponential amplification of the target sequence [9,11,13]. The reaction is carried out under an optimal temperature ranging between 37 and 42 °C with a specially designed primer of 30–35 bases in length, or a normal PCR primer. RPA can amplify targets up to 1.5 kb but is better suited to amplicons between 100 and 200 bp, with an optimal amplification time of 20 min [9,11,13]. Recently, RPA has successfully targeted *Burkholderia* spp., including *Burkholderia pseudomallei* [14,15,16], *Burkholderia mallei* [17], and BCC [18]. More recently, Fu et al. [18] reported a recombinase-aided amplification (RAA) targeting the 16S rRNA gene for BCC detection. However, due to proximity between BCC and other *Burkholderia* species, 16S rRNA gene analysis has limited taxonomic resolution. For instance, it was established that multilocus sequence typing (MLST) (e.g., *atpD*: ATP synthase beta chain; *gltB*: glutamate synthase large subunit; *gyrB*: DNA gyrase subunit B; *recA*: recombinase A; *lepA*: GTP binding protein; *phaC*: acetoacetyl-CoA reductase; and *trpB*: tryptophan synthase subunit B) as a means to identify BCC at the species level surpassed 16S rRNA [4,19]. Therefore, with the availability of whole genome sequences of BCC, interest in a housekeeping gene can provide better resolution in BCC identification. In order to identify all possible BCC-specific PCR primers and exo probes, we used a pan-genome-based bioinformatics pipeline [12].

Finally, we emphasize the development of technology that can rapidly detect and differentiate live from dead BCC cells, which is necessary to reduce the frequency of outbreaks caused by BCC contamination. However, RPA, as so many other isothermal amplification techniques, is unable to separate DNA originating from live or dead cells. Dead bacteria do not cause serious diseases; rather, their presence leads to false positive results and inflated BCC counts [7]. Therefore, we incorporated a sample treatment using propodium monoazide (PMAxx) to overcome this shortcoming [20]. PMAxx is a DNA intercalating dye which penetrates membranes of damaged cells, forms covalent bonds upon exposure to bright visible light, and inhibits PCR amplification. In this study, we aimed to (1) select a specific primer–probe set targeting unique genes; (2) optimize the PMAxx-RPA exo assay for the detection of live BCC cells; and (3) validate the assay in various concentrations of chlorhexidine gluconate (CHX) and benzalkonium chloride (BZK) samples.

## 2. Materials and Methods

### 2.1. Primer and Probe Design

In previous studies, we identified 206 BCC-specific gene clusters which are only present in BCC genomes from a total of 174,715 clusters (i.e., orthologous gene groups with ≥75% sequence identity) of 266 complete genomes (i.e., 82 of BCC and 184 of non-BCC, respectively) of the genus *Burkholderia* [12,20,21]. In this study, among the 206 BCC-specific clusters, we chose a gene cluster (No. of cluster: 3558) encoding a glycine betaine/L-proline transport system permease protein ProW (TC 3.A.1.12.1) (*gbpT*) with three conserved sequence regions of ≥26 bp (Appendix A). The primers and probe were designed according to the manufacturer’s specifications of the TwistDx™ RPA exo kit (TwistDX, Cambridge, UK, www.twistdx.co.uk/docs/default-source/RPA-assay-design (accessed on 3 April 2023)) [22]. The standard parameters of RPA primer design were taken into account and in silico specificity was considered using primer-BLAST software (www.ncbi.nlm.nih.gov). The probe contained a basic nucleotide analog, which replaced a nucleotide in the target sequence flanked by a dT-fluorophore and a corresponding dT-quencher group. These probes were blocked from any potential polymerase extension by a suitable 30–modification group. In a double-stranded context, the tetrahydrofuran (THF) residue presented a substrate for the DNA-repair enzyme exonuclease III, which cleaved the probe at the THF position, thereby separating the fluorophore and the quencher, thus generating a fluorescent signal (see Table 1 for a complete description of the primers and probes for each assay).

### 2.2. 10 μM PMAxx-RPA Exo Assay

PMAxx was utilized as described previously [20]. Among the 24 BCC strains, we chose *B. cenocepacia* J2315 for the PMAxx-RPA exo assay. Briefly, 20 mM PMAxx (Biotium, Fremont, CA, USA) was diluted in high purity water, creating a 1 mM PMAxx stock solution. Then, 10 µL of 1 mM PMAxx was added to 990 µL of a suspension containing live and dead *B. cenocepacia* J2315 (1.5 × 10^8^ colony forming units (CFU)/mL) cells, to a final concentration of 10 μM. Dead cells were obtained from heating 1 mL of suspended cells at 100 °C for 10 min [20]. Following an incubation period of 5 min in the dark, which allowed PMAxx to penetrate dead cells, samples were exposed to blue LED light (PMA-Lite; LED Photolysis Device, Biotium E90002) for 5 min. The samples were then centrifuged at 15,000 rpm for 10 min at 4 °C and the cell pellet washed twice in 1× phosphate-buffered saline (PBS) buffer. The final washed cell pellets were then resuspended in 500 μL of nuclease-free water, 100 μL of that suspension was transferred to a clean 1.5 mL Eppendorf microcentrifuge tube. Total genomic DNA was extracted using the DNeasy UltraClean Microbial Kit (Qiagen, Valencia, CA, USA) and the quantity of extracted genomic DNA was assessed using a NanoDrop ND-2000 spectrophotometer (Thermo Fisher Scientific Inc., Waltham, MA, USA) [7,12,20].

The PMAxx-RPA exo assay was performed in a 50 µL volume using enzyme pellets from the TwistAmp exo kit (TwistDx, Cambridge, UK). The reaction consisted of 29.5 µL rehydration buffer, 2.1 µL forward and reverse primers (10 μM), a 0.6 µL probe (10 μM) and 9.2 µL nuclease-free water. The mixture was briefly vortexed, spun and transferred into a PCR reaction tube. Then, 2.5 µL of magnesium acetate (280 mM) and 4 µL of a DNA template were pipetted into the tube lid. The lid was then closed, the tube was vortexed, spun shortly and immediately incubated under set conditions. The PMAxx-RPA exo assay was performed in a CFX96™ qPCR instrument (BioRad, Hercules, CA, USA) with fluorescence detection in the 6-carboxy-fluorescein (FAM) channel for 2 min at 40 °C followed by 30 cycles for 17 s (20 min). RPA products were then directly analyzed with both a CFX96™ qPCR instrument (relative fluorescence units (RFU)) and 2% agarose gel electrophoresis. Before electrophoresis, exonuclease within the RPA product was inactivated at 95 °C for 5 min [23]. 

### 2.3. Optimization of Temperature, Reaction Time and Concentration of Magnesium Acetate for PMAxx-RPA Exo Assay

To determine the optimal amplification temperature, the PMAxx-RPA exo assay was carried out with 4 µL of approximately 1 ng/µL genomic DNA of *B. cenocepacia* J2315 at different temperatures, namely 35, 37, 40, 42, and 45 °C for a period of 20 min, per manufacturer recommendations. Following optimal temperature selection, the assay was performed for 10, 15, 20 and 25 min reaction times (10–50 cycles in a CFX96™ qPCR instrument) to monitor amplification kinetics. Furthermore, 12, 14 and 16 mM magnesium acetate (280 mM stock solution) was utilized in the optimization process. 

### 2.4. True-Negative Rate (Specificity) and Limit of Detection (LOD)

To evaluate the primer/probe specificity, we used 56 bacterial strains (38 species), which included 13 BCC, 8 non-BCC and 17 non-*Burkholderia* species. The University of Michigan’s *Burkholderia cepacia* Research Laboratory and Repository provided 36 strains of *Burkholderia* spp. and two strains of *Caballeronia* spp. All strains were cultivated as described previously [7]. The total genomic DNA was extracted as described above and adjusted to 10 ng/µL. Then, 4 µL of sample DNA was used in triplicate for the PMAxx-RPA exo specificity assay. The formulas used to calculate the estimated specificity were as described previously [12].

Approximately 94.3 ng/µL of genomic DNA from *B. cenocepacia* J2315 (1.5 × 10^8^ CFU/mL) was used as templates for the RPA to determine the limit of detection (LOD). A serial dilution (4 μL) of the DNA, ranging from approximately 10 ng/µL to 100 fg/µL (9.43 ng/µL, 943 pg/µL, 94.3 pg/µL, 9.43 pg/µL, 943 fg/µL, 94.3 fg/µL), corresponding to approximately 10^7^–10^2^ CFU/mL, was amplified by the PMAxx-RPA exo assay to determine the LOD of the assay. All assays were performed in triplicate.

### 2.5. Effect of PMAxx-RPA Exo Assay in Presence of Antiseptics and Cell Lysates

#### 2.5.1. Evaluation of Various Concentrations of CHX and BZK

To investigate the effect of CHX and BZK on the PMAxx-RPA exo assay, DNA was extracted from cells of *B. multivorans* HI2229 (1.0 × 10^9^ CFU/mL), either PMAxx-treated or untreated. For cross-species validation of our PMAxx-RPA exo assay, we chose in the present study to use *B. multivorans* HI2229, which was susceptible at the lowest CHX and BZK concentration (50 µg/mL CHX and 50 µg/mL BZK) [24]. After DNA extraction, CHX (final concentrations: 0, 10, 100 and 200 µg/mL) and BZK (final concentrations: 0, 50, 200 and 500 µg/mL) were directly spiked into the genomic DNA. To serve as a positive control, 0 μg/mL was not spiked with CHX or BZK for DNA samples. 

#### 2.5.2. Comparing DNA Extraction Methods Using the Boiling Method and a Commercial Kit

To determine the most efficient DNA for live/dead cell discrimination with RPA, we compared cell lysates to pure genomic DNA using the boiling method (cell lysates) and a commercially available kit (pure genomic DNA). Briefly, 4 mL of *B. multivorans* HI2229 cells, OD_600nm_ = 0.2 equivalent to 10^9^ CFU/mL, were spiked with 0.4 mL of 100, 1000 and 2000 μg/mL CHX to achieve final concentrations of 10, 100 and 200 μg/mL CHX, respectively. The treated cells were not incubated, but genomic DNA was extracted from cells with or without 10 µM PMAxx-treatment. Following the 10 µM PMAxx-treatment, the cells were washed in 1 × PBS buffer, followed by centrifugation at 15,000 rpm for 10 min. The supernatant was then removed, and the cells were resuspended in 500 μL nuclease-free water. A total of 100 µL was transferred to a clean 1.5 mL Eppendorf microcentrifuge tube and boiled for 10 min at 100 °C followed by spinning at 1000 rpm using an Eppendorf ThermoMixer^®^ C to extract genomic DNA [7]. The remaining 400 µL of resuspended cells was centrifuged and the resulting pellet was the target of DNA extraction using a commercially available kit (DNeasy microbial kit, Qiagen, Valencia, CA, USA) according to the manufacturer’s instructions. In both cases, the DNA extraction from the kit as well as the boiling method, the purity (A260/A280 ratio) and absorbance ratio (A260/A230 ratio) were measured with a NanoDrop ND-2000 spectrophotometer (Thermo Fisher Scientific Inc., Waltham, MA, USA). Both DNA extracts were adjusted at approximately 10 ng/µL and used as a template for RPA analysis.

### 2.6. Detection of Live/Dead B. multivorans HI2229 in CHX and BZK Solutions

To validate our PMAxx-RPA exo assay, 10^9^ CFU/mL of *B. multivorans* HI2229 (OD_600nm_ = 0.2 equivalent) were treated with 100, 1000, 2000 µg/mL CHX and 500, 2000, 5000 µg/mL BZK to achieve final concentrations of 10, 100, 200 µg/mL CHX and 50, 200, 500 µg/mL BZK, respectively. The treated cells were incubated in the dark at room temperature for 24 h. One milliliter of each bacterial suspension was subjected to the PMAxx treatment as described above. The cells were then centrifuged, and the resulting pellet was resuspended in 1× PBS and centrifuged at 15,000 rpm for 10 min at 4 °C. Pellets were resuspended in 500 µL nuclease-free water and 100 µL of that suspension was used for DNA extraction by the boiling method. For control purposes, DNA was also extracted from cells treated with CHX and BZK without PMAxx treatment. Next, 10 ng/µL DNA of both PMAxx-treated and untreated sample, in triplicate, were used for the PMAxx-RPA exo analysis. Statistical analysis was performed via Student’s *t*-test using SigmaPlot vs. 13.0 software (Palo Alto, CA, USA).

## 3. Results

### 3.1. Optimization of the PMAxx-RPA Exo Assay

The PMAxx-RPA exo assay was carried out in accordance with the manufacturer’s instruction with consideration of parameters such as temperature, reaction time and magnesium acetate concentration in order to optimize the assay. The RPA exo assay with gbpT primer and probe successfully amplified 13 BCC species. To optimize the reaction temperature, five different temperatures (i.e., 35, 37, 40, 42 and 45 °C) were evaluated on 0.9 ng/µL genomic DNA of *B. cenocepacia* J2315. As shown in Figure 1a, the specific target fragment was successfully amplified at 35–45 °C, with a higher amount of amplified product observed at 40 °C (403 ± 79.50 RFU) and 42 °C (431.33 ± 25.81 RFU). It is worth noting that there is no significant difference between the RFU values at 40 and 42 °C. Based on this result, for reducing the temperature, we chose 40 °C as the optimal reaction temperature, which is more practicable for use in field application. Following temperature optimization, different reaction times (10, 15, 20 and 25 min) were investigated. At 10 min, no amplification was observed with an average RFU of 0.18 ± 0.13. A positive signal was detected at 15 min (159.67 ± 25.9 RFU) which increased steadily and plateaued between 20 and 25 min (547 ± 60.51 and 655.67 ± 72.92 RFU) (Figure 1b). As few as 15 min were deemed necessary for amplification with an optimal reaction time of 20 min. We then examined three different concentrations of magnesium acetate, namely 12 mM (414 ± 48.82 RFU), 14 mM (486.33 ± 39.14 RFU) and 16 mM (408.67 ± 17.24 RFU) (Figure 1c). All three magnesium acetate concentrations were successful in amplification with a relatively better fluorescence at 14 mM of magnesium acetate. Thus, the optimal RPA exo reaction parameters were 40–42 °C, 20 min and 14 mM magnesium acetate, which were applied throughout the rest of this study.

### 3.2. Evaluation of Live/Dead Cells with PMAxx Treatment

To determine whether the same PMAxx treatment reported earlier could work with RPA exo [20], we tested both live and dead *B. cenocepacia* J2315 (1.5 × 10^8^ CFU/mL) cells in nuclease-free water with 10 µM PMAxx. Total cells were not treated with PMAxx on live BCC cells and were compared to live cells for examining any possible interference of PMAxx. All the DNA samples were normalized to approximately 1 ng/μL. No statistical difference was observed between total cells (RPA exo assay, 284 ± 82.17 RFU) and live cells (PMAxx-RPA exo assay, 288.4 ± 75.8 RFU). No RFU values were detected from dead cells treated with PMAxx (1.41 ± 0.268 RFU) (Figure 2). Therefore, 10 µM PMAxx was used in the remaining RPA exo assay. 

### 3.3. Specificity and LOD of PMAxx-RPA Exo Assay

To investigate the specificity of the newly designed primer–probe, genomic DNA from 13 BCC species, 8 non-BCC, and 17 non-*Burkholderia* species were tested using the RPA exo assay. The result shows that all 13 BCC (24 BCC strains) yielded RFU (100% sensitivity; 13 positives out of 13), whereas non-*Burkholderia* strains did not yield any RFU (Table 2). Two previously classified *Burkholderia* species which were recently reclassified as *Caballeronia zhejiangensis AU10475,* and *Caballeronia zhejiangensis AU12096* tested negative. However, five species of non-BCC, namely *Burkholderia glumae, Burkholderia gladioli, Burkholderia plantarii, Burkholderia thailandensis* and *Burkholderia oklahomensis ES0634* tested positive. Based on these results, the specificity of design primer probe was 80% ((Number of true negative events (TN)/(Number of false positive events (FP) + TN) × 100; (20 (TN; 20 negatives out of 25)/(5 (FP; 5 false positives out of 25) + 20 (TN)) × 100 = 80))

The LOD of the PMAxx-RPA exo assay was evaluated by assessing a 10-fold serially diluted genomic DNA of *B. cenocepacia* J2315 (approx. 10 ng/µL (corresponding to 10^7^ CFU/mL), 1 ng/µL, 100 pg/µL, 10 pg/µL, 1 pg/µL, 100 fg/µL). Figure 3 shows that the primers set for the *gbpT* gene could detect 10 pg/µL DNA molecules. Analysis by gel electrophoresis showed a band for the PMAxx-RPA exo assay products (lanes with ≥10 pg/µL) (Appendix A). However, the 1 pg/µL (4.4 ± 8.6 RFU) and 100 fg/µL (1.3 ± 0.836 RFU) did not yield any band on agarose gel (Appendix A). Therefore, any value above 50 RFU consistently detected on 2% agarose gel, was considered as a positive reaction. Based on these results of five independent runs, the PMAxx-RPA exo assay with the *gbpT* primers showed an LOD of approx. 10 pg/μL (50 ± 8.6 RFU), corresponding to 10^4^ CFU/mL. 

### 3.4. Assessment of PMAxx-RPA Exo Conditions

#### 3.4.1. Effect of CHX and BZK on the PMAxx-RPA Exo Assay

To assess the effect of antiseptics on the PMAxx-RPA exo assay, 1 ng/µL genomic DNA of *B. multivorans* HI2229 (1.0 × 10^9^ CFU/mL) cells was used in a 50 μL PMAxx-RPA exo assay. Without CHX in DNA, as a positive control, total cells (without PMAxx; 702.08 ± 49.25 RFU) yielded higher RFU values compared to live cells (with PMAxx; 321.11 ± 20.99 RFU). Similarly, when DNA was spiked with 10 µg/mL CHX, for total cells (without PMAxx) and live cells (with PMAxx), we recorded an average RFU of 398.683 ± 4.85 and 143.15 ± 20.99, respectively. No RFU was detected in 100 and 200 µg/mL CHX, in either PMAxx-treated or untreated cells (Figure 4a). Similar results were obtained in DNA samples spiked with BZK where no RFU was detected in 50, 200 and 500 µg/mL BZK (Figure 4b). The sample without BZK (0 µg/mL BZK), untreated, and PMAxx-treated samples yielded 387.66 ± 61.13 RFU vs. 426.29 RFU. These results indicate a potential interference at higher concentrations of CHX (≥10 µg/mL) and BZK (≥50 µg/mL).

#### 3.4.2. Comparing Pure DNA and Cell Lysates on the PMAxx-RPA Exo Assay

We compared two DNA extraction methods, pure genomic DNA (commercial kit) and cell lysates (boiling method) without purification procedure to determine the most suitable and efficient procedure for live/dead differentiation with the PMAxx-RPA exo assay. In total cells (i.e., without PMAxx) with various CHX concentrations (Figure 5a), RFU values from a commercial kit (981–1112 RFU) were much higher than with the boiling method (371–551 RFU). Comparative results were obtained in DNA extracted from a commercial kit for live cells (with PMAxx; 917–1063 RFU) (Figure 5b). However, in live cells (i.e., with PMAxx), much lower amplification rates were recorded in DNA extracted by the boiling method (137–176 RFU) compared to the control (482.65 RFU) (Figure 5b). The commercial kit method did not show any significant difference (*p* > 0.05) in various concentrations of CHX. Although the boiling method showed lower RFU compared to the commercial kit method, the boiling method had sufficient detection capability of PMAxx-RPA exo assay for total or live cells (>50 RFU).

### 3.5. Assessing Live/Dead Cells in CHX and BZK Solutions

The PMAxx-RPA exo assay was applied to approx. 10^9^ CFU/mL of *B. multivorans* HI2229 cultured for 24 h with 0, 10, 100 and 200 µg/mL CHX and 0, 50, 200 and 500 µg/mL BZK. Using the boiling method for DNA extraction, cell lysates were templates for the PMAxx-RPA exo assay. When not treated with CHX, total *B. multivorans* HI2229 (i.e., without PMAxx) showed 297.89 ± 93.88 RFU while the live control (i.e., with PMAxx) yielded 382 ± 206.05 RFU. With 10, 100 and 200 µg/mL CHX, total cells showed no differences with an average of 297.04 ± 52.65 RFU, 418.79 ± 139.91 RFU, and 310.16 ± 45.77 RFU, respectively. Similar results were also obtained from the live cells (i.e., with PMAxx) within the range of 374.54 ± 177.12 RFU, 464.46 ± 116.74 RFU in 10–100 µg/mL CHX. However, when the 200 µg/mL CHX were assayed, total cells showed 310.16 ± 45.77 RFU and live cells decreased to 129.71 ± 70.01 RFU (Figure 6a). The PMAxx treatment was efficient in eliminating signals from dead cells in 200 µg/mL CHX. 

Three BZK treatments were also examined by PMAxx-RPA exo to separate DNA originating from live and dead cells. The RFU values of total cells (i.e., without PMAxx) from 0 µg/mL BZK (596.48 ± 204.48 RFU) and 50 µg/mL BZK (684.53 ± 30.45 RFU) were notably different from those that received 200 µg/mL and 500 µg/mL BZK (281.61 ± 111.87 RFU and 207.82 ± 57.08 RFU). However, the live cells (i.e., with PMAxx) from 0 µg/mL BZK (347.15 ± 148.62 RFU) and 50 µg/mL BZK (459.27 ± 103.24 RFU) displayed similar RFU values of total cells (i.e., without PMAxx). With 200 and 500 µg/mL BZK, RFU values of live cells (130.40 ± 43.08 RFU and 191.47 ± 117.52 RFU) were lower than from total cells (281.61 ± 111.87 RFU and 207.82 ± 57.08 RFU) (Figure 6b). The PMAxx-RPA exo assay, especially in 50 and 200 µg/mL BZK, showed effectiveness in eliminating signals of dead cells, hence providing live/dead cell discrimination.

## 4. Discussion

Rapid and accurate detection of live BCC cells is important for regulatory purposes and essential in protecting public health. To detect BCC cells, conventional PCR, real-time quantitative PCR (qPCR) [2,7,8], droplet digital PCR (ddPCR) [7,20], LAMP [12] and flow cytometry [21] have been developed. Due to its success in rapid detection of many pathogens, the RPA method finds widespread acceptance in low-resource laboratory settings for the detection of pathogens in the field. RPA has a wide range of applications and has been reported as one of the best isothermal techniques for point care diagnostics due to low-cost instrumentation, which is especially suitable for detection in the field or in low-resource settings [9,11,14,15,17]. The ProW(*gbpT*)-based colorimetric PMAxx-RPA exo assay described here can be performed within about 1 h without any enrichment at 40 °C. We observed that after optimization, this assay was able to detect BCC in nuclease-free water and various concentrations of CHX and BZK solutions. 

For the successful detection of BCC using a PMAxx-RPA exo assay, the primers and probe must meet the standard parameters for isothermal amplification. They must also have sufficient biological specificity and sensitivity to allow detection of BCC. For the molecular-based detection of BCC, we previously used two gene clusters encoding a 3,4-dihydroxy-2-butanone 4-phosphate synthase (EC 4.1.99.12)/GTP cyclohydrolase II for LAMP and PMAxx-ddPCR assays [12,20] and an inner membrane protein, KefB/KefC family for the flow cytometry method [21]. These gene clusters have four completely conserved sequence regions of at least 20 bp. The four completely conserved sequence regions in two gene clusters were suitable for either primers or a probe, but not both. In this study, we utilized another gene cluster that encodes a glycine betaine/L-proline transport system permease protein ProW (TC 3.A.1.12.1)(*gbpT*) that has three conserved sequence regions of at least 26 bp [12,20]. These three completely conserved sequence regions of the ProW cluster meet the standard parameters for RPA primer and probe design. However, lab experimental analysis shows cross reactivity with other *Burkholderia* species resulting in 80% specificity. This represents a major obstacle in the correct identification of BCC using one molecular target from closely related non-BCC species as reported previously [3,4,19,25,26,27].

The selected primers were suitable for the PMAxx-RPA exo assay as they successfully amplified selected target genes with amplification products around 383 bp verified via agarose gel electrophoresis and RFU. Since exo probes can result in the exonuclease mediated degradation of DNA, thus digesting RPA reaction final product [9], in this study, we relied on RFU values used for real time-RPA in a positive detection. Therefore, based on our sensitivity analysis, a cutoff value of 50 RFU, which generates a positive signal on gel electrophoresis, was chosen as positive detection. Furthermore, this variable RFU in RPA is due to real-time PCR instrument design. ROX (carboxy-X-rhodamine) passive reference dye can provide a constant fluorescent signal for sample normalization throughout the RPA assay [28]. As previously stated, the LOD and turnaround time vary depending on the RPA assay, based on the target sequence, amplicon size, and the kind of biological sample analyzed, which could all affect RPA efficiency [9,13]. The developed RPA exo assay can detect as little as 10 pg/μL of genomic DNA corresponding to 10^4^ CFU/mL. Similar results were obtained from artificially spiked human blood and urine samples, where the RPA-lateral flow (LF) assay’s detection limit was 4.8 × 10^4^ and 4.95 × 10^4^ CFU/mL of *B. pseudomallei*, respectively [16]. Alternatively, these sensitivity results were slightly lower in comparison to the LAMP assay for detecting 1 pg/µL for *Burkholderia cepacia* complex in non-sterile pharmaceutical products [12]. Although, the established PMAxx-RPA exo assay method provides low sensitivity in detecting live BCC cells from nuclease-free water, this method is advantageous as far as applications in the field, and presumptive tests in facilities with low resources are concerned.

Amplification of DNA from dead cells would provide an unreliable indication of human health risk assessment posed by the presence of pathogens, since the copy number would be falsely increased and thus, would lead to overestimating bacterial counts [20]. Although the RPA assay can detect *B. mallei, B. pseudomallei* and BCC [14,15,16,17,18], it is not effective in distinguishing dead bacteria from live ones. Recently, the PMAxx-RPA assay was successfully employed in the detection of *Salmonella* from spiked milk samples [29]. In this study, compared with the suspension of live bacteria, the suspension of PMAxx-treated dead bacteria showed 1.14 RFU, but the live bacteria showed 288 RFU. As expected, the total bacteria (i.e., without PMAxx) also showed 284 RFU, which means that no PMAxx interference was observed in the RPA exo assay. As we have previously reported on ddPCR, a 10 µM PMAxx and 5 min blue LED light exposure treatment can eliminate signals originating from dead cells [20]. PMAxx combined with RPA exo assay process could also eliminate false positive results from dead bacteria.

Several PCR inhibitors have been shown to be tolerable by RPA, including hemoglobin (20–50 g/L), heparin (0.5 U), urine (up to 5%), and ethanol (4% *v*/*v*), which had no discernible effect on the reaction [9,10,30]. However, the presence of SDS (0.05% *v*/*v*), urine (10% *v*/*v*), and selenite cysteine inhibited RPA [30,31]. In this study, concentrations above 10 µg/mL CHX and 50 µg/mL BZK in genomic DNA interfered with the inhibitory amplification signals with or without PMAxx on the RPA exo assay. However, concentrations ranging 10–200 µg/mL CHX and 50–500 µg/mL BZK solutions did not affect the RPA assay when CHX and BZK-treated cells and genomic DNA were extracted. Our results are consistent with earlier findings where we demonstrated that concentrations ranging 50–100 µg/mL CHX and 10–200 µg/mL BZK did not affect the ddPCR assay amplification [20]. Therefore, for a successful RPA assay in the presence of CHX and BZK, cells should be washed prior to DNA extraction. Furthermore, RPA as well as other PCR based detection assays are susceptible to inhibit and strongly influence the quality of the target DNA preparation to be amplified. For that reason, we evaluated the extraction of DNA using the boiling method and compared the results with that of a commercial kit. The kit extraction method offers better DNA quality with higher RFU recorded, compared to the boiling method. However, the procedure involves multiple steps, which can be labor-intensive, time-consuming, costly and are susceptible to contamination with DNA from other sources [32]. In contrast, the boiling method, which heats a sample up to 100 °C for 10 min, was sufficient to release the target DNA and certify the presence of the BCC [7,12]. This boiling method does not require the purification of DNA from samples. Even though it provides less DNA concentration and quality, the boiling method was sufficient in differentiating total cells (i.e., without PMAxx) and live cells (i.e., with PMAxx) by RPA exo assay. Therefore, omission of the DNA purification step before RPA amplification made the assay less time consuming and less expensive than other available molecular tests.

BZK and CHX are present in a variety of everyday nonsterile products (e.g., mouthwash solutions, hand hygiene lotions, ophthalmic and optic products, multidose aqueous nasal spray) [2,24]. Typically, BZK and CHX are normally applied to those products at concentrations ranging from 0.02% (200 μg/mL) to 5% (50,000 μg/mL), which might not be adequate to eliminate resistant strains. BCC outbreaks can be traced to contaminated CHX and BZK products. Previously, we reported how *B. multivorans* HI2229 was susceptible to 50 µg/mL CHX and 50 µg/mL BZK [24]. Consequently, if disinfectant treatments above 50 µg/mL significantly reduce cell viability and if PMAxx is capable of discriminately removing DNA originating from dead cells, then significant differences or no signal amplification should be detected in treated cells when compared to untreated ones. As expected, cells not treated with PMAxx (total cells) yielded a higher RFU than PMAxx-treated cells (live cells) as indicated by the lower fluorescence observed in PMAxx-treated cells, which was the direct result of elimination of signals emanating from dead cells. These results agree with previous studies demonstrating MIC values of *B. multivorans* HI2229 environmental isolates (soil enriched with anthranilate) exposed to CHX and BZK [24]. Interestingly, in 200–500 µg/mL BZK solutions without PMAxx (total cells), amplification signals were lower than in 0–50 µg/mL BZK solutions. This is most likely due to the mechanism of BZK which is known to disrupt cell membranes [33], consequently releasing DNA which would be lost during the washing step. However, 100 µg/mL CHX and 500 μg/mL BZK concentrations did not provide clear evidence of live/dead assessment, suggesting that either the washing step was insufficient in removing the antiseptics, or higher concentrations of PMAxx may be required to be effective in alleviating high concentrations of antiseptics. Counts of PMA-qPCR for disinfectant-treated *Legionella* in biofilm displayed large variability between replicates [34]. Therefore, Taylor et al., suggest that a higher concentration of PMA might be needed to compensate for the presence of other compounds in an environmental sample [34]. However, alternatively high concentrations of PMAxx may potentially be toxic to BCC, as we have reported earlier [20]. Furthermore, including more washing steps would lead to the loss of cells, yielding unreliable results [35]. Despite the concern with antiseptic interference, in this study, PMAxx-RPA exo assay still recorded live signals, suggesting that the effectivity of PMAxx can enhance live/dead assay performance in non-sterilized pharmaceutical products.

## 5. Conclusions

Here, we reported an RPA exo assay combined with PMAxx for the detection of live *Burkholderia* cells. This assay is quick, simple and offers a new potential way of detecting BCC. The PMAxx-RPA exo assay does not require any sophisticated instrumentations and could be carried out using common lab equipment, such as a simple water bath or portable thermal equipment. The direct boiling method could be used for non-sterilize pharmaceutical samples at a significantly reduced cost to process samples. Compared with the traditional detection method of BCC, the PMAxx-RPA exo assay has the advantages of short detection time and simple operation, making it useful as a presumptive test to detect BCC.

## Figures and Tables

**Figure 1 microorganisms-11-01401-f001:**
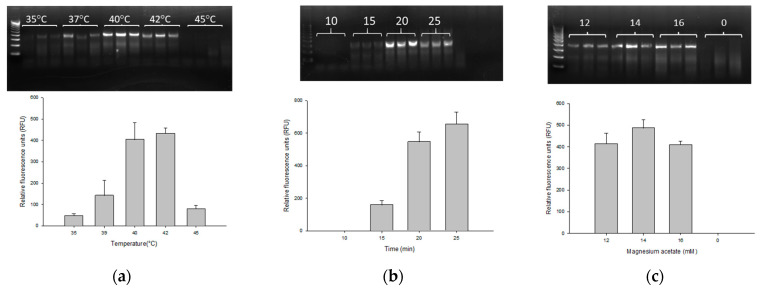
Optimization of temperature (**a**), reaction time (**b**), and concentration of magnesium acetate (**c**) for the PMAxx-RPA exo assay using 0.9 ng/µL of B. cenocepacia J2315. RPA products were directly analyzed with the relative fluorescence units (RFU) by CFX96™ qPCR instrument. Results were confirmed in triplicate of technical replicates by 2% agarose gel electrophoresis. A reaction was considered negative when the RFU value was below 50.

**Figure 2 microorganisms-11-01401-f002:**
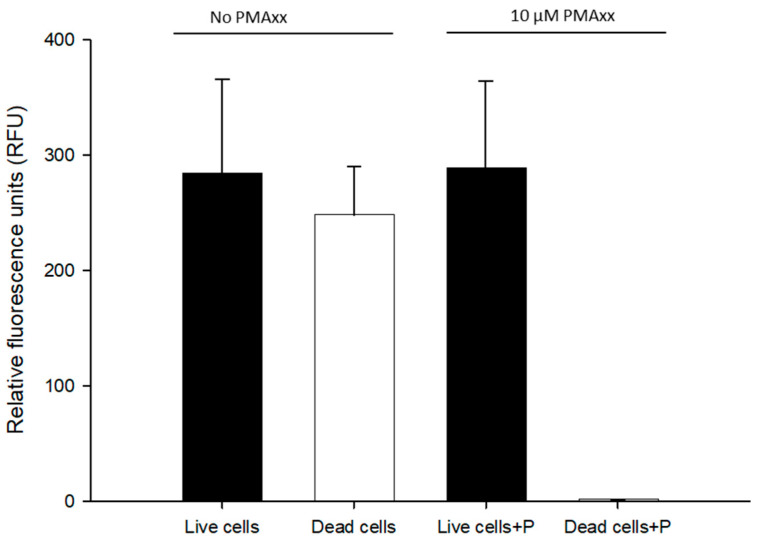
Assessment of 10 µM propidium monoazide (PMAxx) on live/dead *B. cenocepacia* J2315 (1.5 × 10^8^ CFU/mL) cells in nuclease-free water. Live/dead cells were left untreated (RPA exo assay) and live/dead cells+P were treated with 10 µM PMAxx (PMAxx-RPA exo assay). Dead cells were obtained from heating at 100 °C for 10 min and treated with 10 µM PMAxx.

**Figure 3 microorganisms-11-01401-f003:**
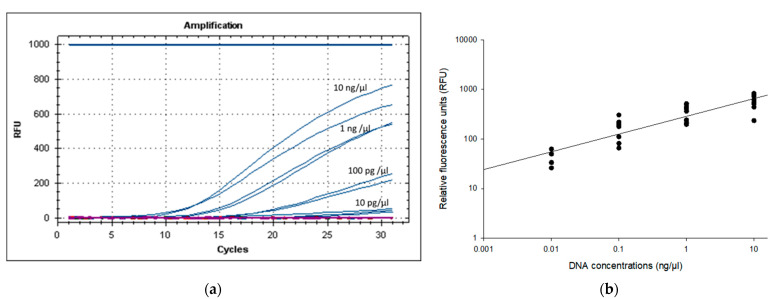
Limit of detection (LOD) of the PMAxx-RPA exo assay (**a**) RFU development and (**b**) scatter plot of RFU versus DNA concentrations (r^2^ = 0.979, y = 88.553ln(x) + 440.2) using 10-fold serial diluted genomic DNA (approx. 10 ng/µL, 1 ng/µL, 100 pg/µL, 10 pg/µL) of *B. cenocepacia* J2315. DNA samples (94.3 ng/μL) were spectrophotometrically determined at A260.

**Figure 4 microorganisms-11-01401-f004:**
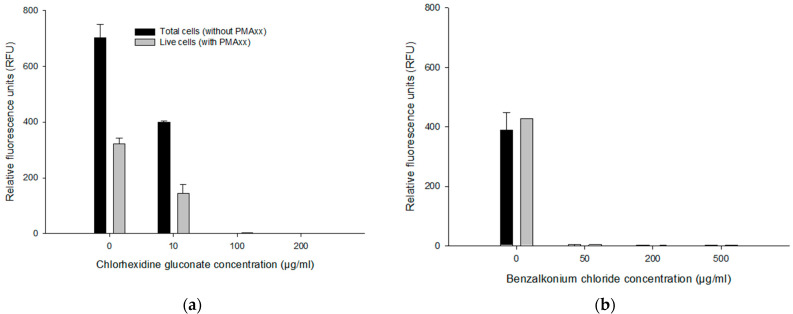
Effects of concentration of CHX and BZK on RPA exo assay. RFU from DNA extract spiked with 10, 100 and 200 µg/mL CHX (**a**), and 50, 200 and 500 µg/mL BZK (**b**). Error bars in diagrams represent the mean value ± standard deviations obtained in triplicate.

**Figure 5 microorganisms-11-01401-f005:**
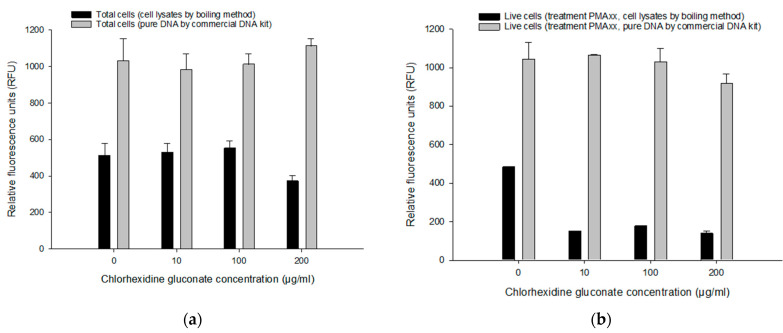
Comparison of cell lysates by boiling DNA extraction method to a commercial kit from different CHX concentrations. RFU from total cells (i.e., without PMAxx) (**a**) and live cells (i.e., with PMAxx) (**b**) treated with 5, 10, 50, 100 µg/mL CHX. Error bars in diagrams represent the mean value ± standard deviations obtained in triplicate.

**Figure 6 microorganisms-11-01401-f006:**
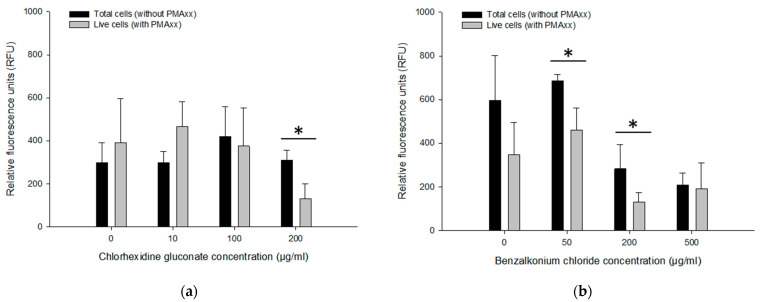
RFU obtained by PMAxx-RPA exo in live *B. multivorans* HI2229 cells in suspension spiked with different concentrations of CHX and BZK. Cells treated with 5, 10, 50, 100 µg/mL CHX (**a**), and 10, 50, 100, 200 µg/mL BZK (**b**). Error bars in diagrams represent the mean value ± standard deviations obtained in triplicate. The * indicates a statistically significant difference of *p* < 0.05.

**Table 1 microorganisms-11-01401-t001:** Primer and probe sequences for the PMAxx-RPA exo assay.

Primer/Probe	Sequence (5′–3′)	Primer Length (nt)
gbpT-F gbpT-R	ACGCTGTCGTCGACGATCATCAGCCTCGTGCT ACCATCGACAGCGCCATCATGATCGTCTGGTT	32 32
Probe	TCGGCCGCGTGCCGGGGATCCTGTCGACGG[FAM-dT]G[THF]-[BHQ1-dT]CTTCGCGATGCCGCC	49

Abbreviations: F, forward primer; R, reverse primer; FAM-dT, fluorescein amidites deoxythymidine; THF, tetrahydrofuran abasic–site mimic; BHQ1-dT, black hole quencher-1 deoxythymidine.

**Table 2 microorganisms-11-01401-t002:** Specificity analysis using 13 BCC species (24 strains), 8 non-BCC (15 strains), and 17 non-*Burkholderia* species.

Group	Species	Strain	Results
BCC	*Burkholderia cepacia*	PC783	+
		AU24442	+
	*Burkholderia stabilis*	AU23340	+
	*Burkholderia ambifaria*	HI2468	+
	*Burkholderia anthina*	HI2738	+
	*Burkholderia metallica*	AU0553	+
		AU16697	+
	*Burkholderia contaminans*	HI3429	+
		AU24637	+
	*Burkholderia diffusa*	AU1075	+
	*Burkholderia arboris*	ES0263a	+
		AU22095	+
	*Burkholderia lata*	HI4002	+
	*Burkholderia multivorans*	HI2229	+
		AU24571	+
	*Burkholderia vietnamiensis*	HI2212	+
		AU24694	+
	*Burkholderia cenocepacia*	AU1054	+
		AU0222	+
		AU19236	+
		HI2976	+
		HI2485	+
		J2315	+
Non-BCC	*Burkholderia glumae*	AU6208	+
		AU12450	+
	*Burkholderia gladioli*	AU26454	+
		AU29541	+
		AU30473	+
		AU16341	+
	*Burkholderia concitans*	AU12121	–
	*Burkholderia oklahomensis*	ES0634	+
	*Burkholderia plantarii*	AU9801	+
		AU37486	+
	*Burkholderia thailandensis*	AU13555	+
		AU36262	+
	*Burkholderia tropica*	AU15822	–
		AU19944	–
	*Burkholderia fungorum*	AU18377	–
		AU35949	–
Non-*Burkholderia*	*Caballeronia zhejiangensis*	AU10475	–
		AU12096	–
	*Enterococcus faecalis*	ATCC29212	–
	*Enterococcus durans*	ATCC6056	–
	*Proteus mirabilis*	ATCC7002	–
	*Enterococcus faecium*	ATCC35667	–
		ATCC49624	–
	*Bacillus subtilis*	ATCC6051	–
	*Citrobacter freundii*	ATCC8090	–
	*Pseudomonas aeruginosa*	PAO1	–
		ATCC27853	–
	*Yersinia enterocolitica* subsp. *entrocolitica*	ATCC27729	–
	*Shigella sonnei*	ATCC9290	–
	*Lactobacillus salivarius* subsp. *salivarius*	ATCC11741	–
	*Enterobacter aerogenes*	ATCC13048	–
	*Klebsiella pneumoniae*	ATCC13883	–
	*Candida albicans* (Robin) Berkhout	ATCC10231	–
	*Salmonella enterica*	isolates	–
	*Paenibacillus lautus*	isolates	–
	*Brevibacillus laterosporus*	isolates	–

+ positive reaction, – negative reaction.

## Data Availability

The data presented in this study are available on request from the corresponding author.

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
