# Peer review of "Propidium Monoazide (PMAxx)-Recombinase Polymerase Amplification Exo (RPA Exo) Assay for Rapid Detection of Burkholderia cepacia Complex in Chlorhexidine Gluconate (CHX) and Benzalkonium Chloride (BZK) Solutions"

_microorganisms, 2023, doi:10.3390/microorganisms11061401_

Round 1

Reviewer 1 Report

In the manuscript entitled “Propidium Monoazide (PMAxx)-Recombinase Polymerase Amplification exo (RPA exo) Assay for Rapid Detection of Viable Burkholderia cepacia complex from Non-Sterilized Pharmaceutical Products”, the authors accessed to an exo probe-based recombinase polymerase amplification with propidium monoazide for rapid and selective detection of viable and nonviable BCC bacteria, which have emerged as one of the most reported contaminants of non-sterile pharmaceutical products. In a previous study, the authors optimized conditions for a PMAxx-Droplet Digital PCR (ddPCR) to detect viable BCC bacteria with the same purpose.  Although the implementation of the PMAxx-RPA exo assay can bring some advantages, namely the reduction of equipment requirements, there are some issues that need to be clarified and I am not entirely convinced of the wide applicability of the described methodology. 

Major Comments:

1.       In the manuscript, among the 206 Bcc specific clusters mentioned, only one was chosen to test and optimize this detection methodology. What are the selection criteria? Why have not you selected and tested other primers and probe, considering the lack of specificity of this choice ( 9 out of 15 non Bcc strains were also positive)?

2.       Line 269: In my opinion a control is missing. Did you test the dead cells without PMAxx and get a positive result?

3.       Why have you started the optimization of the PMAxx-RPA exo assay with B. cenocepacia J2315 and assess the effect of antiseptics on the assay using B. multivorans HI2229?

Minor Comments:

1.       Line 34: “Specifically, Burkholderia cepacia complex, or BCC, is a member of that group that has recently been linked to numerous outbreaks…” should be replaced by “Specifically, bacteria of the Burkholderia cepacia complex, or BCC, are a group of bacteria of this genus that have recently been linked to numerous outbreaks…”

2.       Line 132: The legend of Table 1 is missing.

3.       Line 253: In figure 1 it is not described what are the three wells of each condition in the agarose gels. Are they technical replicates?  

Author Response

Thank you for the valuable comment!

In the manuscript entitled “Propidium Monoazide (PMAxx)-Recombinase Polymerase Amplification exo (RPA exo) Assay for Rapid Detection of Viable Burkholderia cepacia complex from Non-Sterilized Pharmaceutical Products”, the authors accessed to an exo probe-based recombinase polymerase amplification with propidium monoazide for rapid and selective detection of viable and nonviable BCC bacteria, which have emerged as one of the most reported contaminants of non-sterile pharmaceutical products. In a previous study, the authors optimized conditions for a PMAxx-Droplet Digital PCR (ddPCR) to detect viable BCC bacteria with the same purpose.  Although the implementation of the PMAxx-RPA exo assay can bring some advantages, namely the reduction of equipment requirements, there are some issues that need to be clarified and I am not entirely convinced of the wide applicability of the described methodology. 

Major Comments:

  1. In the manuscript, among the 206 Bcc specific clusters mentioned, only one was chosen to test and optimize this detection methodology. What are the selection criteria? Why have not you selected and tested other primers and probe, considering the lack of specificity of this choice ( 9 out of 15 non Bcc strains were also positive)?

Thank you for the valuable comment! Based on the text that appears on lines 126-129, “In previous studies, we identified 206 BCC-specific gene clusters which are only present in BCC genomes from a total of 174,715 clusters (i.e., orthologous gene groups with ≥75% sequence identity) of 266 complete genomes (i.e., 82 of BCC and 184 of non-BCC, respectively) of the genus Burkholderia [12,20,21].”

For the successful detection of BCC using a PMAxx-RPA exo assay, the primers and probe must meet the standard parameters for isothermal amplification. Also, for the molecular-based detection of BCC, we previously used two gene clusters encoding a 3,4-dihydroxy-2-butanone 4-phosphate synthase (EC 4.1.99.12)/GTP cyclohydrolase II for loop-mediated isothermal amplification (LAMP) and PMAxx- droplet digital PCR (ddPCR) assays and an inner membrane protein, KefB/KefC family for the flow cytometry method (Daddy Gaoh et al., 2021; 2022).

As mentioned in the Discussion section (Line 414-420),These gene clusters have four completely conserved sequence regions of at least 20 bp. The four completely conserved sequence regions in two gene clusters were suitable for either primers or a probe, but not both. In this study, we utilized another gene cluster that encodes a glycine betaine/L-proline transport system permease protein ProW (TC 3.A.1.12.1)(gbpT) that has three conserved sequence regions of at least 26 bp [12,20]. These three completely conserved sequence regions of the ProW cluster meet the standard parameters for RPA primer and probe design.” Therefore, we selected those primers for full lab analysis. We feel that we have already addressed the question of the reviewer.

  1. Line 269: In my opinion a control is missing. Did you test the dead cells without PMAxx and get a positive result?

We agree that inclusion of an additional control (i.e., dead cells without PMAxx) would be helpful. To clarify, we have added “dead cells without PMAxx” in Figure 2.

  1. Why have you started the optimization of the PMAxx-RPA exo assay with B. cenocepaciaJ2315 and assess the effect of antiseptics on the assay using B. multivorans HI2229?

B. cenocepacia  was susceptible at the highest CHX concentrations (100-500 μg/ml) and BZK concentrations (50-500 μg/ml). The antiseptics MIC values of B. cenocepacia were over the tested ranges with 10-200 µg/mL CHX and 50- 500 µg/mL BZK. However, as mentioned in the Discussion section (Line 497-498), “. Previously, we reported how B. multivorans HI2229 was susceptible to 50 µg/mL CHX and 50 µg/mL BZK [24].” Therefore, we used B. multivorans HI2229 to know whether the proportions for live/dead are different among values of antiseptics treatment (i.e., control vs. under or over MIC antiseptic treatment). For the sake of clarity, we added “For cross-species validation of our PMAxx-RPA exo assay, we chose in the present study to use B. multivorans HI2229 which was susceptible at the lowest CHX and BZK concentration (50 µg/mL CHX and 50 µg/mL BZK) [24].” (Line 209-211).

Minor Comments:

  1. Line 34: “Specifically, Burkholderia cepaciacomplex, or BCC, is a member of that group that has recently been linked to numerous outbreaks…” should be replaced by “Specifically, bacteria of the Burkholderia cepacia complex, or BCC, are a group of bacteria of this genus that have recently been linked to numerous outbreaks…”

For the sake of clarity, we have changed the expression “Specifically, Burkholderia cepacia complex, or BCC, is a member of that group that has recently been linked to numerous outbreaks” into the expression “Specifically, strains of the Burkholderia cepacia complex, or BCC, consist of a group of bacteria that have recently been linked to numerous outbreaks.” (Line 35-38). 

  1. Line 132: The legend of Table 1 is missing.

Thank you for pointing this out! We have added the following: “Table 1. Primer and probe sequences for the PMAxx-RPA exo assay.”

  1. Line 253: In figure 1 it is not described what are the three wells of each condition in the agarose gels. Are they technical replicates?  

For the sake of clarity, we have added “Results were confirmed in triplicate of technical replicates by 2% agarose gel electrophoresis.” (Line 277-278).

Reviewer 2 Report

Authors reported a newly developed exo probe-based RPA method for the quantitative determination of live/dead BCC cells in chlorhexidine gluconate (CHX) and benzalkonium chloride (BZK) matrices. This method exhibited high sensitivity and selectivity. The employment of PMAxx to prohibit the interference of dead cells is promising. In general, the experiments were well designed and processed. The manuscript was well written and smooth. So, I recommend the editor to accept this manuscript in present form.

Author Response

Thank you for your kind comments regarding the novelty of our study! 

Reviewer 3 Report

The manuscript describes the validation of a method for detection of Burkholderia cepacia complex in non-sterilized pharmaceutical products using a RPA assay in combination with PMA xx, as detection system. Overall the manuscript is well written, the results are clearly presented. One question: did the authors check the sensitivity and specificity of the RPA assay using an established real-time PCR protocol? Also, the RPA exo-assay without PMA xx, as detection system, was applied to non-BCC strains?

Thank you

Author Response

We appreciate giving us opportunity to address these comments. 

Comments and Suggestions for Authors

The manuscript describes the validation of a method for detection of Burkholderia cepacia complex in non-sterilized pharmaceutical products using a RPA assay in combination with PMA xx, as detection system. Overall the manuscript is well written, the results are clearly presented. One question: did the authors check the sensitivity and specificity of the RPA assay using an established real-time PCR protocol?

We appreciate giving us opportunity to address these comments. We didn`t use real-time PCR. Because these primers were not suitable for real-time PCR because the exo RPA assay required 30 to 35 nucleotides long primer with GC content of between 40% and 60%. In addition to a TwistAmp® exo probe which should be 46-52 nucleotides long, with at least 30 of which are placed 5’ to the THF site, and at least a further 15 are located 3’ to it (TwistDX, Cambridge, UK, www.twistdx. co.uk/docs/default-source/RPA-assay-design).

Also, the RPA exo-assay without PMA xx, as detection system, was applied to non-BCC strains?
To investigate the specificity of the newly designed primer/probe, genomic DNA from 13 BCC species, 8 non-BCC, and 17 non-Burkholderia species were tested using the RPA exo assay. The result shows that all 13 BCC (24 BCC strains) yielded RFU (100% sensitivity; 13 positives out of 13), whereas non-Burkholderia strains did not yield any RFU (Table 2).

Thank you

Reviewer 4 Report

Comments to the Authors:  

The manuscript entitled “Propidium Monoazide (PMAxx)-Recombinase Polymerase Amplification exo (RPA exo) Assay for Rapid Detection of Viable Burkholderia cepacia complex from Non-Sterilized Pharmaceutical Products” developed a PMAxx-RPA method to rapidly detect viable Burkholderia cepacia complex. The authors optimized the PMAxx-RPA exo assay for the detection of live BCC cells with a self-designed probe and then validated the assay in various concentrations of chlorhexidine gluconate (CHX) and benzalkonium chloride (BZK) samples. However, the experimental design and the novelty is poor, and the manuscript organization is not satisfying. There are many mistakes in the paper writing as well as the result explanation. Many conclusions are not sufficiently supported by the results.

Major:

1.        The title is “Propidium Monoazide (PMAxx)-Recombinase Polymerase Amplification exo (RPA exo) Assay for Rapid Detection of Viable Burkholderia cepacia complex from Non-Sterilized Pharmaceutical Products”. However, the authors did not detect the viable BCC in real samples, which is strongly recommended to be investigated.

2.        Please explain the difference between “viable” and “live”. Then choose only one word in the manuscript.

3.        Line 18. The target of RPA is DNA, rather than protein.

4.        Line 19. Detection limit of 10 pg/μL of WHAT?

5.        Please explain why CHX and BZK are investigated as “Non-Sterilized Pharmaceutical Products” is the authors’ concern.

6.        There are many spelling and grammar mistakes and long sentences I would not list. Please polish the manuscript.

7.        Line 65-72. The authors have developed a LAMP method with a detection limit of 1.034 pg/μL. Why did the authors have to develop the PMAxx-RPA method? The authors listed that the shortcoming of LAMP is “relies on a complex primer design at an optimal temperature of 65 °C”. It is NOT the shortcoming to establish a LAMP method to detect BCC because the primers have been identified by your group! These sentences in the manuscript are very confusing. Rewrite here to explain the necessity to develop the PMAxx-RPA method.

8.        Line 73. The principle of the RPA method is redundant here. Try to shorten these sentences and explain “exo”.

9.        Line 107. The authors declare that they selected a specific primer-probe set targeting unique genes. However, no related research is found in the manuscript. The gbpT is directly used here without any selection.

10.    Line 139. Why do the authors use B. cenocepacia J2315 here? What is the relationship between B. cenocepacia J2315 and Burkholderia cepacia complex? Make it clear in the manuscript.

11.    Line 164. CARELESSNESS. Rewrite the section title.

12.    Line 191. Explain why to use another new species, B. multivorans HI2229, here.

13.    Again, why use CHX and BZK? Are CHX and BZK normally applied in non-sterilized pharmaceutical products?

14.    Section 2.5.2. Why only use CHX here? Is the result different when using BZK?

15.    Section 2.6. B. multivorans HI2229, instead of B. cenocepacia J2315 was used. Explain this. The incubation time of CHX and BZK on the cells is 24 h. Are these conditions (time and concentration) consistent with the actual situations? Any references?

16.    Figure 1A. The SD in 40℃ is wider than 42℃, which means the detection precision of 42℃ is higher. I suggest the authors reconsider the optimal temperature. Figure 1B. “Plateaued between 20 and 25 min”? 30 and 35 min should also be analyzed to draw this conclusion.

17.    Figure 2. The experimental design is poor. The authors should prepare different proportions of viable and dead BCC and testify whether the method accurately identifies these viable/dead proportions.

18.    Line 284-285. The calculation is confusing. Try to make it clearer.

19.    Figure 4A&B. Why the proportion of viable/total cells is different when no CHX and BZK were added? The “0” point should be the same.

20.    Figure 5. Why did the authors choose the boiling method? Because “the boiling method used for DNA extraction, appears to depend on CHX concentration.”?? It cannot support the conclusion, and it DID NOT depend on CHX concentration. The traditional culture-based method MUST be performed to verify whether CHX and BZK efficiently kill BCC. The traditional method results should be used to compare with the PMA method.

21.    Figure 6B. “0” point, nearly 30% of cells are dead? The results are not persuasive at all because the viability of the cells is not normalized. Again, the traditional method should be used for comparison.

22.    The real non-sterilized pharmaceutical products should be tested with the method.

23.    Line 381. 20 min. The pretreatment time should be involved.

24.    Line 414. “However” is nonsense here. Line 414-425. Invalid discussion and comparison here. The merit of the developed method is not exhibited at all.

25.    Again, why did the authors test the influence of CHX and BZK? Are they widely used in real samples? The authors must explain this.

There are many mistakes in the paper writing as well as the result explanation. 

Author Response

Thank you for the valuable comment! Please refer to the attachment.
